# Public Health Awareness on Bat Rabies among Bat Handlers and Persons Residing near Bat Roosts in Makurdi, Nigeria

**DOI:** 10.3390/pathogens11090975

**Published:** 2022-08-26

**Authors:** Veronica Odinya Ameh, George J. Chirima, Melvyn Quan, Claude Sabeta

**Affiliations:** 1Department of Veterinary Public Health and Preventive Medicine, College of Veterinary Medicine, Federal University of Agriculture Makurdi, Makurdi 2373, Nigeria; 2Agricultural Research Council, Soil Climate and Water, 600 Belvedere Street, Arcadia, Pretoria 0001, South Africa; 3Department of Geography, Geoinformatics and Meteorology, University of Pretoria, Lynwood Road, Hatfield, Pretoria 0028, South Africa; 4Department of Veterinary Tropical Diseases, Faculty of Veterinary Science, University of Pretoria, P Bag X04, Onderstepoort 0110, South Africa; 5WOAH Rabies Reference Laboratory, Agricultural Research Council, Onderstepoort 0110, Pretoria 0002, South Africa

**Keywords:** bats, viruses, rabies, knowledge, association

## Abstract

Rabies is a neglected disease endemic in Asia and Africa but is still a significant public and veterinary health threat. Whilst a key delicacy for the local diet, bats are a natural reservoir host for many viral zoonotic agents including lyssaviruses, the causative agent of rabies. Studies on knowledge and practices linked to the disease will help to identify gaps and define preventive strategies that may subsequently result in a reduction and the potential elimination of human rabies. In order to assess the public health awareness of bat rabies among specific population groups in Makurdi (Nigeria), structured questionnaires (n = 154) were administered by face-to-face interviews to bat handlers and persons residing near bat roost sites. A total of 59.7% of the respondents were persons residing near bat roost sites, 13% were bat hunters, 25.3% were bat meat consumers and 1.9% were university researchers. Only 6.5% of respondents reported using some form of personal protective equipment (PPE) ranging from hand gloves, face/nose masks and protective boots to lab coats/coveralls while handling bats, whilst the majority (93.5%) did not use any form of PPE. With a mean knowledge score of 8.34 out of a possible 12 points, 50.6% of respondents had good knowledge of bats and their disease-carrying potential, 39.6% had fair knowledge, while 9.7% had poor knowledge. Log linear models showed significant associations between knowledge score and level of education, as well as knowledge score and occupation. The latter highlights the requirement to enhance public education among bat handlers and persons residing near bat roosts on the need to protect themselves better, while handling bats particularly during processing of bats for food and on steps to take when exposed to bites from bats.

## 1. Introduction

Rabies is an acute form of viral encephalomyelitis, which is almost invariably fatal and affects all mammalian vertebrates globally, with the exception of some island nations, the Antarctica and some European countries such as the United Kingdom, Switzerland, Germany, France, Finland, Belgium, Luxembourg, Italy and Netherlands amongst others [1,2,3]. Transmission of this highly neurotropic virus occurs normally through bites with infectious saliva, but also through scratches or via skin lesions [3]. The aetiological agent of rabies is a member of the order *Mononegavirales*, family *Rhabdoviridae* and genus *Lyssavirus*, that currently consists of 17 genetically and antigenically related species [4]. Rabies virus (RABV) is generally associated with the infection of terrestrial carnivores and 15 of the *Lyssavirus* species, with the exception of *Lyssavirus mokola* (MOKV) and *Lyssavirus ikoma* (IKOV), have been isolated from bat species. MOKV and IKOV have both been recovered from terrestrial species in Africa with the former having been isolated on numerous occasions from rodents, dogs, and cats, and the latter is a single isolation from a rabid African civet (*Civettictis civetta*) from Tanzania [5]. More recently, *Matlo bat lyssavirus* (MBLV), a potentially new species was identified from an apparently healthy Natal long-fingered bat (*Miniopterus natalensis*) in the northern parts of South Africa and shown to be very closely related to West Caucasian bat virus (WCBV) in phylogroup III [6]. Similar to LBV and MOKV, current rabies vaccines are unlikely to protect against this recently identified lyssavirus. RABV is the most important member of the *Lyssavirus* genus, given that it is responsible for at least 59 000 human rabies deaths annually, mostly in Africa and Asia [7,8,9]. Bats are mammals in the order Chiroptera with their forelimbs adapted as wings, they are the only mammals naturally capable of true and sustained flight. After rodents, they are the largest order, making up about 20% of mammalian species [10]. The order Chiroptera has been divided into 2 suborders, Yinpterochiroptera and Yangochiroptera, based on phylogenetic analyses [11,12]. The smallest bats can weigh as little as 2–2.6 g, while the largest bats (flying foxes) can weigh as much as 1.6 kg. Yinpterochiroptera (Pteropodiformes) includes members of the super family Pteropodidea and five othert families: Rhinopomatidae, Rhinolophidae, Hipposideridae, Craseonycteridae, and Megadermatidae which make up the super family Rhinolophoidea. The suborder Yangochiroptera (Vespertilioniformes) is made up exclusively of species that use laryngeal echolocation [13]. Pteropodidea eat fruits, nectar, or pollen, while the Rhinolophoidea and members of the suborder Yangochiroptera are insectivorous, and others feed on fruit, nectar, pollen, fish, frogs, small mammals, or blood [14].

Several studies have shown exposure of *E. helvum* to RABV and other lyssavirus species, with the potential of transmitting viral pathogens to man and other mammalian carnivores [15,16,17,18,19,20,21]. There is paucity of information on the epidemiology of bat rabies in Nigeria as most individuals find it difficult to submit sick bats to veterinary laboratory for diagnosis as these bats are usually associated with bad omens, witchcraft and supernatural powers in Nigeria [21].Viral pathogens that can potentially be transmitted from bat species to humans include Lyssaviruses [22,23,24], Coronaviridae [25,26,27], Paramyxoviridae [28,29,30], Filoviruses [24,31], Influenza viruses [32] and Reoviruses [33,34]. Bats are a special delicacy and an important source of protein for some people in West Africa including parts of Nigeria [35,36,37]. There is little knowledge among bat consumers on how to limit the ability of this diet source to pass on pathogens to humans especially during the processing of bat meat. Consequently, strategies to fight rabies are impaired or constrained. In areas where bat meat is consumed, the bats are sold in secluded areas/spots called “Joints”. The meat is usually prepared by grilling/roasting or boiling with hot spices and sold to consumers in these “Joints” along with alcoholic beverages. Bats are not generally recognized as a source of food/meat in Nigeria considering that less than 1% of Nigerians consume them. However, the demand for bat meat has resulted in constant interactions between humans and the bat populations thereby posing a great public health risk. This study therefore investigated the interactions between bat handlers, persons residing near bat roosts and their knowledge on bat rabies and the disease-carrying and -transmitting potential of bats with a view to enhancing the fight against rabies and achieving health for all by 2030 Sustainable Development Goals (SDG 3).

## 2. Methods and Materials 

### 2.1. Study Sites

This study focused on geographical areas in Makurdi, Benue State, Nigeria (Figure 1). Makurdi is the capital of Benue State, a town located in the North Central part of Nigeria, along the Benue River (7◦44025.700 N 8 ◦31052.800 E). The town is divided by the Benue River into the northern and southern banks, which are connected by two bridges, the railway bridge built in 1932 and the new dual carriage bridge commissioned in 1978. Owing to its location in the valley of the River Benue, Makurdi experiences warm temperatures for most of the year. The period from November to January, when the Harmattan weather is experienced, is, however, relatively cool. Agriculture forms the backbone of the Benue State economy, and this industry engages more than 70% of the working population. Makurdi has a large bat population, with these bats roosting on trees in and around the Benue State Government House (in Makurdi) and on trees in private residences close to the government house (7°44′25.7″ N 8°31′52.8″ E). Roost sizes vary between 100,000 to about a million bats depending on the season of the year. Bat hunting is a big business in Makurdi, with some of the hunters exporting bat meat from Makurdi to the Eastern parts of Nigeria where there is a huge demand. The meat is sold in ‘Joints’ to locals and consumed together with locally brewed alcohol. The bats in this study are the straw-coloured fruit bats (*Eidolon helvum*) that roost on trees in residential areas and around the Government House in Makurdi.

### 2.2. Methods

A cross-sectional study was designed to assess bat handlers and people residing near bat roosts on their awareness of the rabies-carrying potential of bats and how bats can impact public health. A simple random sampling technique was applied. A total of (390) invitations were sent out but only 154 accepted and participated in the surveys. Structured questionnaires were prepared (see Appendix A), pretested (n = 20) and administered by face-to-face interviews (n = 154) to bat handlers and persons residing near bat roost sites. The questionnaires were administered between October 2019 to December 2020 exclusively to individuals that were available and had given their consent to participate in this study.

### 2.3. Data Analysis

The data were analyzed using the Statistical Package for Social Sciences, SPSS (Version 26, SPSS Inc. Chicago, IL, USA). To assess the level of knowledge of the respondents, a marking scheme containing expected answers was prepared and used to score the responses. One point was assigned for each correct answer and a zero for an incorrect answer. The maximum scores that could be obtained for knowledge was 12 points. We did an exploratory data analysis following [39] and then used a frequency distribution curve to come up with our categories. The scores were categorized such that respondents scoring between 0–4 points (0–33%) were considered as having poor knowledge, 5–8 points (40–65%) as fair and 9–12 points (>65%) were considered as a good knowledge score or high level of awareness. First, to explore and describe simple and straight forward associations between demographic variables and the categorized scores, the χ^2^ test of association was applied. Values of *p* < 0.05 were considered significant in the χ^2^ analysis. To improve statistical sensitivity with investigating complex associations including investigating the effects of several interactions as well as additive effects of the variables, the log-linear modelling technique, which is a more advanced framework for analyzing categorical data and has advantages because of its ability to handle interactions or combinations of variables was applied [40]. We used Log linear models in combination with Akaike’s Information Criteria (AIC) as described by Burnham and Anderson [41] to compare the relative weight of support by the data to models including their interactions [42,43]. Delta AIC (∆_i_) and Akaike weights (*w_i_*) were also calculated and used to assess model support by the data [41]. Models are ranked using AIC. The model with the least AIC values is the most parsimonious [41].

## 3. Results

### 3.1. Demographic Characteristics of Respondents

The majority of respondents were males (74.7%) in the 20–30 age group (55.2%) as shown in Table 1. Ninety-four (69.0%) of respondents were married, while the rest (31.0%) were single. Forty-six (29.9%) of the respondents were civil servants, 31.2% were businessmen/women, 9.1% farmers, 7.8% were hunters and 22.1% were unemployed (Table 1).

### 3.2. Association with Bats

The association of respondents with bats is shown in Table 2, with 100% admitting to having had contact with bats during their lifetime. Only 6.5% of the respondents reported using some form of PPE ranging from hand gloves, face/nose masks, protective boots to lab coats/coveralls while handling bats, while the majority (93.5%) did not use any PPE at all during handling the bats. Persons residing near bat roosts accounted for 59.7% of respondents, 25.3% were bat meat consumers, while 13.0% of the respondents were bat hunters. Twenty (13.0%) of the respondents reported being exposed to bites from bats and 35% of those exposed to bat bites did not get any form of treatment for the bites or post-exposure management (Table 2).

### 3.3. Association of Demographic Variables with Categorized Knowledge Scores

Most of the respondents (50.6%) had good knowledge of bats and their disease-carrying potential, 39.6% had fair knowledge, while 9.7% had poor knowledge. There was no statistically significant association between knowledge score and age, gender or marital status of respondents. There were evident and significant associations between knowledge score and level of education, knowledge score and occupation with *p*-values of 0.01 and 0.028, respectively (Table 3).

### 3.4. Knowledge of Respondents on Rabies and Disease-Carrying Potential of Bats

The mean knowledge score was 8.34 out of 12 points. The majority of respondents (96.1%) had heard of rabies and correctly indicated bites from infected animals (81.8%) as the most common mode of rabies transmission. One hundred and twenty-eight of the respondents (83.1%) concurred that bats carry pathogens, while only 59.1% believed that these bat pathogens could be transmitted to humans (Table 4).

### 3.5. Effects of Demographic Variables on Knowledge Score

The model with the lowest value of AIC is the best-supported model and exhibits the best fit from a set of candidate models [44]. From the data in this study, the best fitting model for single factor influence of demographic variables on knowledge score is the ‘level of education’, with an AIC value of −1270.44 (Table 5). The model with additive effects of age and marital status, age and level of education, gender and level of education received substantial statistical support, with AIC values of −1376.24, −1355.59 and −1307.64, respectively (Table 5).

The top four ranked models in this study are age and marital status, age and level of education, gender and level of education, and occupation and level of education. The first two models are strongly supported and the last two weakly supported by the data (Table 5).

## 4. Discussion

This study was undertaken with a view to evaluate the interface between bat handlers, persons residing near bat roosts in Makurdi, and their knowledge on bat disease-carrying potential for possible risk of transmission of bat rabies. It is important to note that the knowledge gaps identified amongst the residents of the study area can be potentially useful for the prevention of this neglected zoonosis.

Of all the demographic variables, effects of level of education and effect of occupation had the most positive influence on the level of knowledge of respondents on bat disease-carrying potential. This is aligned with findings from previous studies that investigated knowledge, attitudes, and practices to canine rabies among dog owners in Wukari, Nigeria [45]. The respondents with tertiary level education and civil servants had better knowledge of rabies and in addition, adopted better preventive measures against the disease in the Ameh study [45]. Al-Mustapha et al., [46] also obtained similar results, albeit that they showed very low awareness and knowledge of canine rabies among residents of Kwara state, Nigeria [46]. In Tanzania, Sambo et al. [47] found out that people who had higher level of education, originated from areas with a history of rabies interventions, had experienced exposure to a suspect rabid animal, were male and owned dogs, were more likely to have greater knowledge about canine rabies [47].

The top ranked models in this study comprise age, level of education, occupation and gender. This means that these variables are influential on the level of knowledge of the respondents on bat pathogen-carrying potential. This is in consonance with work done by Sambo et al. in Tanzania, where he found that males had better knowledge of canine rabies than females. This might be attributed to the fact that males generally were better educated than the females involved in this study. It also lends support to studies carried out by Almustapha and Ameh, where they found that respondents with tertiary level education and civil servants had better knowledge of canine rabies [45,46].

Age is an important variable in this study and appears in the best-supported model. It is therefore possible that as one gets older, one tends to acquire more knowledge through exposure to daily life activities and consequently may have more experience. This in turn makes the individual better equipped to handle bats.

Presently, it is well known that insectivorous, frugivorous and hematophagous bats in the Americas act as wildlife reservoirs for lyssaviruses and can transmit the disease to humans [19]. The first record of a rabid insectivorous bat in Europe occurred in Hamburg, Germany, in 1954 [48]. Since this account, there have been other confirmed human rabies cases arising through infection with European bat lyssaviruses and other unconfirmed reports [19]. Fifteen of the seventeen lyssavirus species have exclusively been found in bat species [5,6,49,50], with most associated with rabies in humans and spill-over to terrestrial mammals such as the water-mongoose (Atilax paludinosus) in South Africa [51].

It is therefore a concern that some individuals do not seek proper medical treatment when exposed to bites and scratches from bats. Thirteen percent of the respondents in this study reported being exposed to bat bites (Table 2), which can serve as a potential means of exposure to bat-borne pathogens including rabies and 35% of these eposures were not managed at all. The exposure data are similar to those obtained in studies carried out by Si [52]. In his studies, he found that there were 1515 potential exposures to *Australian bat lyssavirus* (ABLV) in Queensland (Australia) between 2009–2014 [52]. The majority (96%) of these potential exposures were as a result of contacts with Australian bats (members of the family *Pteropodidae* and the species *Saccolaimus flaviventris)*, which are responsible for maintaining and transmitting ABLV to humans and animals via bites, scratches or contact of saliva with mucous membrane/broken skin. ABLV has been documented to cause fatal disease in humans and has been detected in Australian bats including four species of Pteropodidae (all flying foxes) and the *Saccolaimus flaviventris* (yellow-bellied sheath tail bat) [52].

Vora et al. carried out a study on exposure to bat lyssaviruses among humans in areas that commemorate bat festivals in Nigeria [53]. In the Vora study, although >50% of bats had neutralizing antibodies to *Lyssavirus lagos bat*, *Lyssavirus shimoni*, *Lyssavirus mokola* and *Lyssavirus ikoma*, no human participant had detectable neutralizing antibodies to the four lyssaviruses [53]. Other independent studies [16,17,18,21,36] have amply demonstrated previous exposures to four LBV lineages in *Eidolon helvum*, suggesting that *Eidolon helvum* can be a potential source of infection to humans and other animals.

However, evidence generated thus far raises pressing questions about human health risks. The potential implications of bat rabies are particularly salient in Southeast Asia because human–bat interactions occur routinely in many localities. For instance, in Thailand, bat guano is regularly mined from caves for use as a fertilizer. Furthermore, hunting of bats for sale and personal consumption is commonly practiced, despite the existence of laws to stop this practice. The presence of large numbers of bats at many Buddhist temples also facilitates exposures, as these sites are focal points for commerce, tourism, and religious expression. In a survey undertaken in Thailand amongst persons at risk for bat exposure, Robertson [54] found that although general awareness of rabies transmission and severity were relatively high, specific awareness of bat rabies in particular was low, with only 10% of participants identifying bats as a potential source of rabies. In the same study, 36% of the persons failed to say they would take any specific action if bitten or scratched by a bat [54]. This observation is similar to results obtained in this study (see Table 2), where only 15% of respondents who had been exposed to bat bites sought medical care. The majority of those exposed did not seek medical care (35%), while (50%) treated the bite wounds on their own, without seeking medical advice. Contrary to a study by Robertson et al., 2011, respondents (59.1%) in our study recognized bats as sources of pathogens, with the majority (79, 2%) admitting that bats are a source of rabies to both humans and animals (Table 4). The results from our study are in direct contrast to those obtained in a study conducted in two communities in Guatemala [55] to determine the knowledge, attitudes, and practices of persons towards bats and rabies. Despite the risk of rabies posed by hematophagous bats, an overwhelming majority of the respondents (90%) reported that they knew little or nothing about rabies.

## 5. Conclusions

The findings from this study show that interaction and possible exposure to bats in communities near bat roosts and among bat handlers is common, with respondents with tertiary education and civil servants more knowledgeable on bat disease-carrying potential. The level of education of an individual usually determines what occupation he or she is involved in, and this may have an effect on or influence their interaction with bats and the knowledge they have on bats and their pathogen-carrying potential. There is therefore an urgent need for more public enlightenment or enhanced education among bat handlers and persons residing near bat roosts to raise awareness of bat-associated rabies, which will result in a greater likelihood of instituting better preventive measures and health-seeking behavior. This can be achieved by organizing workshops, seminars, radio programs, and talks on rabies and the dangers of handling bats in the local languages in these communities. This will help to reduce the spread of bat-associated rabies and other zoonoses and achieving health for all by 2030 (SDG 3).

## Figures and Tables

**Figure 1 pathogens-11-00975-f001:**
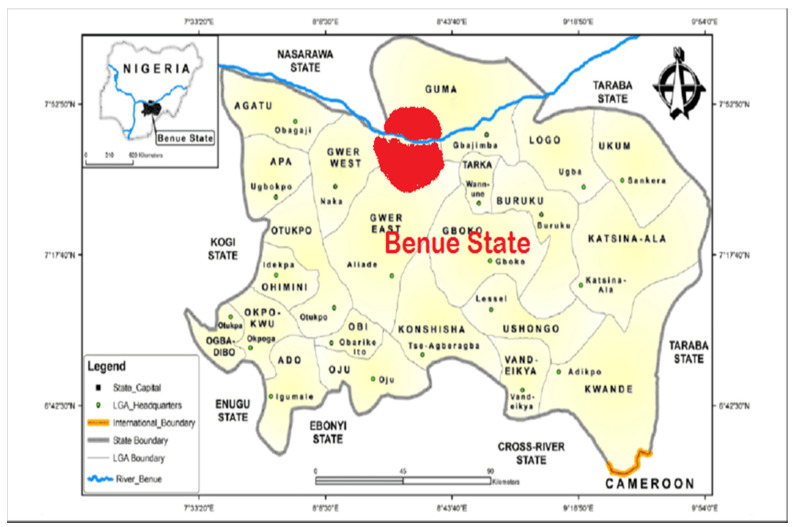
Map of Benue State, showing Makurdi in red [38].

**Table 1 pathogens-11-00975-t001:** Demographic characteristics of respondents in Makurdi.

Variable	Total Number of Respondents (n = 154)	Specific Rates (%)
**Gender**		
Male	115	74.7
Female	39	25.3
**Age**		
<19	15	9.7
20–30	85	55.2
31–40	44	28.6
>40	10	6.5
**Marital Status**		
Single	60	31.0
Married	94	69.0
**Occupation**		
Unemployed	34	22.1
Civil servant	46	29.9
Businessman/woman	48	31.2
Farmer	14	9.1
Hunter	12	7.8
**Level of Education**		
Informal	2	1.3
Primary	13	8.4
Secondary	76	49.4
Tertiary	63	40.9

**Table 2 pathogens-11-00975-t002:** Respondents’ Association with bats.

Type of Association	Total Number of Respondents N = 154	Specific Rates (%)
**Had contact with bats**		
Yes	154	100
No	0	0
**Duration of contact with bats**		
1–5 years	72	46.8
6–10 years	48	31.2
11–15 years	24	15.6
>15 years	10	6.5
**PPE used while handling bats**		
None/use bear hands	144	93.5
Coverall/lab coats	1	0.6
Boots	1	0.6
Hand gloves	7	4.5
Face/nose mask	1	0.6
**Nature of association with bats**		
Bat hunter	20	13.0
Bat meat consumer	39	25.3
Reside near bat roost	92	59.7
Researcher	3	1.9
**No of bat contacts per day**		
<5	32	20.8
5–10	21	13.6
10–15	32	20.8
15–20	50	32.5
>20	19	12.3
**Have you ever been bitten by a bat**		
Yes	20	13.0
No	134	87.0
**Remedy for bat bite**		
Got first aid treatment	1	5.0
Sought medical care	3	15.0
Washed the bite with plenty of soap and water	5	25.0
Got anti-tetanus shot	3	15.0
Used traditional medicine	1	5.0
Did nothing	7	35.0

**Table 3 pathogens-11-00975-t003:** Association of demographic variables of respondents with categorized knowledge scores.

Variable	Categorized Scores N = 154 (%)			*χ^2^*	(df)	*p*-Value
	Poor	Fair	Good			
**Gender**				0.749	1	0.688
**Male**	**12(7.8)**	47(30.5)	56(36.4)			
**Female**	**3(1.9)**	14(9.1)	22(14.3)			
**Age**				5.077	3	0.534
<19	2(1.3)	7(4.5)	6(3.9)			
20–30	9(5.8)	36(23.4)	40(26)			
31–40	4(2.6)	16(10.4)	24(15.6)			
>40	0(0)	2(1.3)	8(5.2)			
**Marital Status**				1.338	1	0.512
Single	7(4.5)	26(16.9)	27(17.5)			
Married	8(5.2)	35(22.7)	51(33.1)			
**Occupation**				17.185	4	0.028 *
Unemployed	3(1.9)	15(9.7)	16(10.4)			
Civil servant	1(0.6)	16(10.4)	29(18.8)			
Businessman/woman	4(2.6)	20(13.0)	24(15.6)			
Farmer	5(3.2)	4(2.6)	5(3.2)			
Hunter	2(1.3)	6(3.9)	4(2.6)			
**Level of Education**				38.538	3	0.000 *
Informal	0(0)	0(0)	2(1.3)			
Primary	6(3.9)	3(1.9)	4(2.6)			
Secondary	8(5.2)	40(26.0)	28(18.2)			
Tertiary	1(0.6)	18(11.7)	44(28.6)			
**Nature of association with bats**				6.861	3	0.334
Bat hunters	2(1.3)	11(7.1)	7(4.5)			
Bat meat consumers	4(2.6)	18(11.7)	17(11.0)			
Reside near bat roost	9(5.8)	32(20.8)	51(33.1)			
Researcher	0(0)	0(0)	3(1.9)			

* Statistically significant *p*-value ≤ 0.05.

**Table 4 pathogens-11-00975-t004:** Assessment of knowledge of respondents on rabies and disease-carrying potential of bats.

Knowledge Item	N = 154	Frequency	(%)
**Have you heard of rabies?**			
Yes		148	96.1
No		3	1.9
I don’t know		3	1.9
**Rabies affects only animals**			
Yes		19	12.3
No		122	79.2
I don’t know		13	8.4
**How is rabies transmitted?**			
Bite		126	81.8
Contact		6	3.9
I don’t know		22	14.3
**Do bats carry disease-causing pathogens?**			
Yes		128	83.1
No		3	1.9
I don’t know		23	14.9
**What pathogens do bats carry?**			
Viruses		68	44.2
Bacteria		23	14.9
Protozoans		8	5.2
Parasites		2	1.3
All of the above		42	27.3
None of the above		11	7.1
**Can bats transmit disease pathogens to other animals?**			
Yes		98	63.6
No		5	3.2
I don’t know		51	33.1
**Can bats transmit disease pathogens to humans?**			
Yes		91	59.1
No		12	7.8
I don’t know		51	33.1
**Can bats transmit rabies to humans and animals?**			
Yes		122	79.2
No		8	5.2
I don’t know		24	15.6
**How do bats transmit diseases to humans?**			
Bites		131	85.1
Contact		7	4.5
I don’t know		16	10.4

**Table 5 pathogens-11-00975-t005:** Effects of demographic variables on knowledge score.

Model ID	Models	AIC	Delta AIC	Akaike Weight (wi)	Parameter Number (k)
1 **	Age + Marital Status + Knowledge score	−1376.24	0	0.99997	7
2	Age + Level of Education + knowledge score	−1355.59	20.651	0.00003	9
3	Gender + Level of Education + Knowledge score	−1307.64	68.601	0.00000	7
4	Occupation + Level of Education + knowledge score	−1303.96	72.281	0.00000	10
5	Marital Status + Level of Education + Knowledge score	−1276.01	100.232	0.00000	7
6	Age + Gender + Knowledge score	−1273.42	102.816	0.00000	7
7	Level of Education + Knowledge score	−1270.44	105.801	0.00000	6
8	Age + Occupation + Knowledge score	−1269.74	106.496	0.00000	10
9	Age + Knowledge score	−1236.22	140.016	0.00000	6
10	Occupation + Gender + Knowledge score	−1221.79	154.446	0.00000	8

** Best-supported model.

## Data Availability

The data that support the findings of this study are available on request from the corresponding author. The data are not publicly available due to privacy or ethical restrictions.

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
