# Peer review of "Public Health Awareness on Bat Rabies among Bat Handlers and Persons Residing near Bat Roosts in Makurdi, Nigeria"

_pathogens, 2022, doi:10.3390/pathogens11090975_

Round 1

Reviewer 1 Report

Comments:

1. Ref. 1 looks outdated to describe the current rabies distribution. Many countries in Europe have defeated rabies and are currently rabies-free (not only island nations!).

2. Materials and Methods section must be placed before results, not after!

3. It would be good to know from respondents if they ever had any health issues (not only bites) after contacting bats either laboratory confirmed or not.

4. I suggest to add you questionnaire as supplementary material to this article.

5. Because the article is mostly about rabies, I would suggest to mention rabies in the title.

Reviewer 2 Report

This is an interesting study investiating the level of bat rabies awareness among individuals in direct or inderect contact with bats in Nigeria. The conclusions of the study highlight the importance of the public awareness on bat rabies in the area.

 Major comments

1.      It is reccomended for the authors to reduce the size of the introduction section, focusing mainly on the lyssaviruses species affecting bats, the epidemiological situation of bat rabies in Nigeria, the bat meat consumption habits,  as well as emphasizing on the targets of their research.

2.      In the introduction section, please provide more information on the consumption of bat meat by residents in Nigeria. How this meat is processed before consumption?What is the percentage of population consuming this meat?

3.      Did the authors used a specific statistical methodolgy for the determination of the sample size in their research? (as they describe in the Methods section, they initially sent the invitation to 390 participants but only 154 individuals finally corresponded)

4.      I was not able to see the questionnaire presented in the Appendix 1.

Minor comments

1.      In the Abstract  authors wrote “Ninety-two (54%) instead of Fifty ninety-two

Round 2

Reviewer 1 Report

The authors have corrected the manuscript. It can now be accepted.

Reviewer 2 Report

 The authos have addressed  the reviewers' comments